# The Impact of Strength Changes on Active Function Following Botulinum Neurotoxin-A (BoNT-A): A Systematic Review

**DOI:** 10.3390/toxins17080362

**Published:** 2025-07-23

**Authors:** Renée Gill, Megan Banky, Zonghan Yang, Pablo Medina Mena, Chi Ching Angie Woo, Adam Bryant, John Olver, Elizabeth Moore, Gavin Williams

**Affiliations:** 1Department of Physiotherapy, Epworth Rehabilitation, Epworth Healthcare, Richmond, Melbourne 3121, Australia; megan.banky@epworth.org.au (M.B.); pab.physio@gmail.com (P.M.M.); angiewoo820@gmail.com (C.C.A.W.); jolver@bigpond.net.au (J.O.); elizabeth.moore@epworth.org.au (E.M.); gavin.williams@epworth.org.au (G.W.); 2School of Physiotherapy, The University of Melbourne, Parkville, Melbourne 3010, Australiaalbryant@unimelb.edu.au (A.B.); 3Department of Medicine, Monash University, Clayton, Melbourne 3168, Australia

**Keywords:** botulinum neurotoxin-A, function, strength, spasticity, upper limb, lower limb, activity, participation, quality-of-life

## Abstract

Botulinum neurotoxin-A (BoNT-A) injections are effective in reducing focal limb spasticity; however, their impact on strength and active function needs to be established. This review was a secondary analysis aimed at evaluating changes to active function in the context of muscle strength changes following BoNT-A intramuscular injection for adult upper and lower limb spasticity. The original review searched eight databases (CINAHL, Cochrane Central Register of Controlled Trials (CENTRAL), Embase, Google Scholar, MEDLINE, PEDro, PubMed, Web of Science) and was conducted with methodology that followed the Preferred Reporting Items for Systematic Reviews and Meta-Analyses (PRISMA) guidelines as described in section 6.2 of Gill et al. For this secondary analysis, no databases were searched; only further data were extracted. The current and preceding review were registered in the Prospective Register of Systematic Reviews (PROSPERO: CRD42022315241). Twenty studies were screened for inclusion, and three studies were excluded because active function was not assessed in all participants. Seventeen studies (677 participants) met the inclusion criteria for analysis. Quality was examined using the PEDro scale and modified Downs and Black checklist and rated as fair to good. Pre- and post-BoNT-A injection strength (agonist, antagonist, and global), active function (activity), participation, and quality-of-life outcomes at short-, mid-, and long-term time points were extracted and analysed. Significant heterogeneity and limited responsiveness in strength and active function outcome measures limited the ability to determine whether changes in strength mediate an effect on active function. Further, variability in BoNT-A type and dose, adjunctive therapies provided, and variability in reporting limited analyses. Overall, no clear relationship existed between the change in muscle strength and active function following BoNT-A injections to the upper and lower limbs for focal spasticity in adult-onset neurological conditions.

## 1. Introduction

Adult-onset neurological conditions are a priority health concern worldwide, with significant and increasing associated costs [1,2]. Muscle spasticity, or muscle resistance to passive stretch [3,4,5,6], is a common disabling sequela for people with neurological conditions. Spasticity is prevalent in up to 43% of people post-stroke [7,8], up to 63% post-traumatic brain injury [9], and up to 60% in multiple sclerosis (MS) populations [10,11].

Improved active function (i.e., dressing or walking) for people with neurological conditions may be dependent on multiple factors [12,13], such as reducing spasticity and increasing muscle strength [14,15,16]. The most effective treatments for each of these impairments, as supported by level 1 evidence, are Botulinum Neurotoxin-A (BoNT-A) intramuscular injections to reduce spasticity [17,18,19] and progressive resistance training to improve strength [20,21,22]. Botulinum neurotoxin-A type, dose, and injection practices should adhere to recommended guidelines [23]. However, as shown in animal and paediatric studies, it is possible that BoNT-A may weaken muscles that are important for active function [24,25], which may be a source of concern for injectors [26].

A recent large systematic review found that the impact of BoNT-A on muscle strength was inconclusive, with no clear evidence of weakness [27]. This is contrary to the common belief that BoNT-A injections weaken muscles and may compromise active function [28,29,30]. However, where strength changes did occur, the review did not report the implications of these changes on active function, which is arguably more clinically important [27]. Active function refers to any ability where the participant actively performs a task with the upper or lower limb (such as walking or self-care). Reporting whether post-injection strength changes influence active function is crucial because it may contribute to understanding how BoNT-A injections impact important active function outcomes [31,32]. Yet, this remains a knowledge gap in the literature.

This current systematic review aimed to evaluate the relationship between changes in muscle strength and active function following BoNT-A.

## 2. Results

### 2.1. Studies Included

Figure 1 demonstrates the Preferred Reporting Items for Systematic Reviews and Meta-Analyses (PRISMA) flow diagram of study identification [33]. This flow diagram has been modified to reflect that this review was a further analysis of a systematic review by Gill et al. [27]. A total of 20 studies were included in the data analysis of Gill et al. [27] and therefore screened for eligibility against the inclusion criteria of the present study. After screening, 17 studies [34,35,36,37,38,39,40,41,42,43,44,45,46,47,48,49,50] met this review’s inclusion criteria (Figure 1). Three studies were excluded because they did not include a measure of active function for all participants [51,52,53].

Of the 17 included studies, eight were randomised controlled trials (RCTs) [34,36,37,38,40,41,43,44], and nine were non-RCTs [35,39,42,45,46,47,48,49,50]. Among the eight RCTs, one was double-blinded [40], and seven were single-blinded [34,36,37,38,41,43,44].

Ten studies (59%) examined strength and active function following BoNT-A injections to the lower limb [34,35,36,37,38,39,40,42,48,50], and seven (41%) studies examined strength and active function following BoNT-A injections to the upper limb [41,43,44,45,46,47,49].

### 2.2. Study Characteristics

The 17 studies involved 677 participants who had strength and active function assessed pre- and post-BoNT-A injections. Sample sizes were between 14 and 140 participants, with a median of 20 participants. Thirteen (76%) evaluated stroke, three (18%) studies evaluated hereditary spastic paraplegia, and one (6%) study evaluated incomplete spinal cord injury. Table 1 (Upper Limb) and Table 2 (Lower Limb) summarise the characteristics of the included studies based on the limb injected.

### 2.3. Outcomes: WHO-ICF Framework

All studies (*n* = 17) examined activity (Table 3 and Table 4, Appendix A) [34,35,36,37,38,39,40,41,42,43,44,45,46,47,48,49,50], four studies (24%) examined quality of life (QoL) (Appendix A) [36,41,43,44], and one study examined a participation outcome (Appendix A) [36]. For ease of presentation, results for participation and QoL outcomes were collated together (Appendix A).

For Table 3 and Table 4, the numbers in square brackets show individual study results. Appendix A provides a detailed breakdown of outcome measures used in each study.

### 2.4. Overall Outcome

A large variety of outcome measures were used in the included studies. Twelve different strength outcome measures and 30 active function outcome measures were used, not enabling the completion of a meta-analysis (Table 1 and Table 2) [34,35,36,37,38,39,40,41,42,43,44,45,46,47,48,49,50].

There were 308 matched strength and active function/participation/QoL relationships analysed for this review. This included 159 (52%) outcomes reported less than 6 weeks post-injection, 79 (26%) outcomes between 6 weeks and 3 months, and 70 (23%) outcomes between 3 months and less than 12 months post-injection.

In general, where active function outcomes improved (81/308; 26%), both upper or lower limb strength remained unchanged (61/81; 75%), with a small proportion of increased (12/81; 15%) and decreased strength (8/81; 10%) results [34,35,36,37,38,39,40,41,42,43,44,45,46,47,48,49,50]. Overall, there was no clear relationship between strength and active function outcomes.

Table 3 (upper limb) and Table 4 (lower limb) are summaries of the outcomes, and each of the individual outcomes was reported in detail in Appendix A. Extensive data, including mean, standard deviations, within-group differences, treatment groups, and follow-up time-points, are outlined in Appendix A. It is noted that varying BoNT-A type/products, dosages, and injection methods were used across the same muscles injected, and there was frequent missing data. Botulinum neurotoxin-A type, dose, dilution per muscle group, injection techniques, and adjunctive therapies, where reported, have been supplied in Supplementary File S5 (S5).

#### 2.4.1. Strength and Active Function Outcomes of the Upper Limb (*n* = 7)

Strength (36/53; 68%) and active function outcomes of the upper limb (37/53; 70%) were mostly unchanged across all time points [41,43,44,45,46,47,49]. For example, there was no change in grip strength or active function between 3 and 12 months in the study by Lannin et al. [43]. When changes in upper limb strength did occur, active function was mostly unchanged. In the single study where weakening of the upper limb did occur, there was no deterioration in active function [49].

Lim et al. [46] used the Modified Barthel Index (MBI) to report active function outcome for the upper limb. However, it is noted that only 5 (i.e., personal hygiene, bathing self, feeding, dressing, toileting) of the 10 items are relevant to the upper limb [46,54].

#### 2.4.2. Strength and Active Function Outcomes of the Lower Limb (*n* = 10)

Despite a trend for greater lower limb weakness in the agonists early post-injection (39/49; 80%), active function remained unchanged (36/49; 73%) or improved (13/49; 27%) (Table 4 and Appendix A). Notably, no studies reported greater antagonist weakness or deterioration in active function at any time point (Table 4 and Appendix A) [34,35,36,37,38,39,40,42,48,50]. For example, a study by Rousseaux et al. [48] found that dorsiflexion and plantarflexion strength remained unchanged, with no evidence of worsened active function, as it either improved or remained unchanged [48].

#### 2.4.3. Strength, QoL (*n* = 4) and Participation (*n* = 1) Outcomes

No clear relationships existed between strength changes and QoL or participation following BoNT-A [36,41,43,44]. Where strength and QoL or participation outcomes were reported, strength was primarily unchanged (14/18; 78%), and the QoL and participation outcomes were also largely unchanged (14/18; 78%). Bollens et al. [36] was the only study to assess participation using the SATISPART questionnaire, reporting no change in strength (i.e., dorsiflexion or plantarflexion) or participation outcomes.

### 2.5. Meta-Analysis

Due to the heterogeneity of study designs, strength and active function outcome measures, and assessment time points, a meta-analysis was not warranted.

### 2.6. Quality Assessment

Appendix A presents the total and subscale scores for the modified Downs and Black checklist and Physiotherapy Evidence Database (PEDro) scales (*n* = 17).

For all 17 studies, the modified Downs and Black median total score was 18 (range: 15–23) out of a maximum possible score of 28, indicating overall fair quality evidence that may affect the reliability of findings [55]. Nine (53%) studies included in this review were non-RCTs, which are prone to several biases, including confounding, performance, selection, and observer bias [35,39,42,45,47,48,49,50,56]. Four (24%) studies were missing data and at risk of reporting bias [40,48,49,50]. Most studies satisfied reporting criteria for hypothesis, primary outcomes, patient characteristics, and interventions, with a median score of 10 (range: 8–11) out of a possible 11 for this section of the checklist. External validity scores were poor, with a median of 0 (range: 0–3) out of a possible 3. The median internal validity for confounding and bias subscale scores was 3 (range: 1–7) out of a possible 7 and 5 (range: 0–6) out of a possible 6, respectively. Only one study by Giray et al. [41] reported power calculations.

Additionally, the 8 RCTs demonstrated a median PEDro score of 8 (range: 7–8) out of a possible 10, which indicated good quality evidence [55]. Half of the trials failed the concealed allocation item in this review.

## 3. Discussion

This systematic review found no clear relationship between changes in strength and active function following BoNT-A injections for adults with spasticity. In instances where active function outcomes did improve, most upper and lower limb agonist and antagonist strength outcomes remained unchanged. Overall, active function remained unchanged or improved, with no reported reduction in active function outcomes following BoNT-A injections. This is clinically important because concerns amongst injectors that BoNT-A may cause weakness and consequently worsen active function were not supported [34,35,36,37,38,39,40,41,42,43,44,45,46,47,48,49,50]. Given the “fair” modified Downs and Black scores, results should be interpreted with caution. However, this concern is somewhat mitigated by the “good” PEDro classification assigned to the RCTs.

This review identified several avenues for further investigation to provide clarity on the relationship between strength and active function following BoNT-A injections. First, several issues with the outcome measures were identified. There was considerable heterogeneity in strength (*n* = 12) and active function (*n* = 30) outcome measures, which precluded the completion of a meta-analysis [57,58]. The large proportion of unchanged results may indicate that no significant changes in strength or active function occurred. Alternatively, it may be that the outcome measures used in these studies had limited responsiveness [59], are partly subjective (e.g., MRC), and lack precision [27]. Additionally, some of the active function measures have demonstrated high measurement error in people with spasticity (e.g., BBT, NHPT) [60,61] or floor effects (e.g., timed stair climb, FIM-Lower Limb) [62,63].

Second, we found limited improvement in active function. A possible reason for this may be that improved active function is co-dependent on reduced spasticity and improved strength and motor control. Spasticity guidelines recommend treatments that address the negative features (i.e., strength and motor control) of the upper motor neuron syndrome (UMNS) [64,65,66]. However, given the heterogeneity and limited responsiveness in strength and active function outcome measures, it is not possible at this stage to determine whether changes in strength mediate an effect on active function. Given no clear relationship was identified within this and the preceding review [27], future research could prioritise evaluating the combined effects of BoNT-A injections and physical therapies such as strength training and task practice, as this may be required to achieve improved active function [21,67,68,69].

Third, in response to issues with outcome measures, several spasticity guidelines have recommended using the Goal Attainment Scale (GAS) as an active function outcome measure because it avoids floor effects, has greater responsiveness than standard measures, and evaluates goals that are specific to the patient [30,70,71,72]. Despite this recommendation, the GAS was only used in two studies involving the same cohort [43,44]. Implementing the GAS more often in spasticity research may enable easier comparisons between studies and lead to future meta-analyses involving tailored and meaningful patient goals [73].

Fourth, few studies reported mid- and long-term outcome evaluations (i.e., >6 weeks to 12 months), which prevented a thorough analysis throughout the entire rehabilitative process following BoNT-A injection [74]. It is well known that the effects of BoNT-A begin within a few days following injection [75], peak between 2 and 4 weeks and gradually decline until 3 months post-injection [74], so understandably, a significant proportion of outcomes (52%) were assessed in the short term (i.e., <6 weeks). While a reduction in spasticity may occur quickly, true improvement in muscle strength at a physiological level does not typically occur until after 8–12 weeks of adjunctive strength-based training [27,76,77]. Therefore, short-term evaluations following BoNT-A injection may fail to detect changes in strength and active function [78].

Fifth, as demonstrated in the results, active function may improve despite strength remaining unchanged, suggesting that other neurophysiological impairments may co-occur. The primary aim of BoNT-A injections is to reduce spasticity, and in doing so, this may “unmask” any existing, usable strength, allowing for improved movement. It appears that a change in active function may occur without any significant increase in force-generating capacity. It is possible that active functional gains could also be attributed to changes in other impairments, such as coordination, proprioception, motor control, and/or reduced antagonist co-contraction or unpleasant sensations, which are not captured through strength assessment [79]. Additionally, manual muscle testing has poor discriminative ability, where important changes in strength may occur but are undetectable. For example, MacAvoy et al. [80] demonstrated that among the six possible manual muscle testing grades (i.e., 0–5), grade 4 represented more than 96% of the reported scoring options for muscle strength assessment [80]. Clinicians and injectors should consider the impact of BoNT-A injections on other impairments in addition to strength when aiming for optimal effect on active function.

Lastly, most studies examined stroke, even though spasticity is prevalent in other neurological conditions such as MS and traumatic brain injury [9,10,11]. The relationship between strength and active function may differ between conditions, and this review has highlighted the knowledge gap within the broader neurological population. Other factors such as the age of participants, delayed onset and chronicity of spasticity, other impairments (e.g., cognition), and the type/frequency of adjunctive therapies may also contribute to changes in muscle strength and active function. Botulinum neurotoxin-A type, dose, dilution, and guidance may all impact treatment efficacy. It is noted that varying BoNT-A products, dosages, and injection methods were used for the same muscles injected, and data were often not reported. Therefore, a deeper analysis of the type, dose, and injection methods, such as ultrasound guidance, may be warranted in future studies to assess the need for greater precision in BoNT-A injection techniques to minimise any potential risk of muscle weakening [81]. However, these analyses were outside the scope of this review.

## 4. Conclusions

Overall, there was no clear relationship between a change in muscle strength and active function following BoNT-A injections to the upper and lower limbs for focal spasticity in adult-onset neurological conditions.

## 5. Future Directions

Based on this review’s findings and those of Gill et al. [27], future studies should use precise, responsive, goal-specific outcome measures in the short, mid-, and long-term to identify any relationship between changes in strength and active function. This review emphasises the need for a “core outcome set” suitable for adults with spasticity in future research [82,83,84]. This would include a group of measures that would be recommended and reported in spasticity-related research, similar to those for ischemic heart disease [85], chronic pain [82], or the Stroke Recovery and Rehabilitation Roundtable recommendations [83]. Such an approach would assist in the standardisation of assessment for future research protocols and enable future meta-analyses examining the relationship between changes in muscle strength and active function after BoNT-A [84].

## 6. Methods

### 6.1. Study Design

This is a secondary analysis of a recent systematic review conducted by Gill et al. [27], which aimed to evaluate the effect that BoNT-A injections have on muscle strength in adult-onset neurological conditions. This secondary analysis aimed to examine the relationship between changes in muscle strength and active function following BoNT-A injection. Both systematic reviews were registered in the International Prospective Register of Systematic Reviews (PROSPERO) database, registration number: CRD42022315241. The methods and reporting of results used in both reviews followed the PRISMA guidelines and checklist [27,33].

### 6.2. Search Strategy

The original review conducted by Gill et al. [27] involved a search of eight electronic databases, including CINAHL, Cochrane Central Register of Controlled Trials (CENTRAL), Embase, Google Scholar, MEDLINE, PEDro, PubMed, and Web of Science. The search was limited to studies conducted on humans and published in English. All included databases were searched from inception. Search strategies included medical subject headings (MeSH) and keywords mapped to titles and abstracts of articles utilising database-specific Boolean operators and syntax. Searches were customised for each database as described in Supplementary File S6 of Gill et al. [27].

### 6.3. Eligibility Criteria

Studies selected for further analysis (*n* = 20) in Gill et al. [27] were screened for inclusion in this current systematic review. The following additional inclusion criteria were applied to the current systematic review:A measure of active function, participation, or quality of life was used to assess all participants.The measure of active function was relevant to the muscle injected (agonist) or the opposing muscle (antagonist).The measure of active function demonstrated established clinometric properties (i.e., data related to validity, reliability, etc. have been reported).

For this current review, “active function” referred to any ability where the participant actively performed the task with the upper or lower limb. Measures of “passive function”, e.g., pain, contractures, or skin breakdown [86,87], were excluded from this review.

### 6.4. Study Selection

Two review members (R.G, M.B) independently appraised all full-text articles to determine if they met the inclusion criteria of this current review. Both reviewers reached agreement on whether to include or exclude articles based on the aforementioned inclusion criteria. As required, a third reviewer (G.W) was available to resolve disagreements between judgments. Two reviewers (R.G, M.B) analysed the newly extracted active function (activity), participation, and QoL data from studies that met the inclusion criteria of the present review. When results were extracted and synthesised from the included studies, statistical significance was outlined as *p* ≤ 0.05, unless the study stated otherwise.

### 6.5. Data Extraction

All relevant data from the articles for inclusion were extracted and recorded using customised Microsoft Word (Version 16.77.1) tables and a Microsoft Excel (Version 16.77.1) template by one author (R.G). Additional reviewers (M.B, Y.Z) assessed accuracy and completeness. Data including the author, study design, population, sample size, mean age, and all measures of strength were previously extracted in the original review [27]. Data extracted included all measures of active function, participation, and QoL for all time points. This included the outcome measure(s) used, units of measure, and statistical data such as means, medians, standard deviations, *p*-values, and time points assessed. When data was missing, all authors were contacted via email twice to request additional information and were allowed at least two months for further data provision, with success from 4/12 authors contacted.

### 6.6. Methodological Assessment of Quality

Quality assessments of the included studies in the original review utilised the modified Downs and Black checklist (non-RCTs and RCTs), and PEDro scales (RCTs) remained unchanged for this review [27].

### 6.7. Data Analysis and Synthesis

Data synthesis involved extracting all measures of strength, active function, participation, and QoL at all time points. Further exclusions were made if outcome measures were not specific to function. The active function results were then pooled with the upper and lower limb strength outcomes as reported by Gill et al. [27] and separated according to recommended spasticity guideline follow-up time frames (i.e., 0 to 6 weeks, 6 to 12 weeks, and greater than 3 months to less than or equal to 12 months) and according to the direction of change (i.e., stronger, no change, weaker for strength, or improved, no change, or worsened for active function) [64,88]. The newly extracted active function outcome results were paired with the relevant strength outcomes within each study and tabulated for visual synthesis. In instances where outcome measures within studies had multiple subscales, only the results from subscales relevant to active function were reported. In studies with multiple treatment arms (i.e., control and experimental groups), data were extracted for each arm where all participants had received a BoNT-A injection. Therefore, numerous results from the same study reporting different strength and/or active function outcome results were reported.

Lastly, summary statistics such as frequency (i.e., the number of occurrences) and proportions (i.e., the relative frequency in percentages) were used to assist in interpreting the corresponding strength and active function, participation, and QoL outcomes. For example, the number of outcomes and proportion of active function changes corresponding with strength changes were calculated at each time point. These statistics were collated into tables and provided in Appendix A.

## Figures and Tables

**Figure 1 toxins-17-00362-f001:**
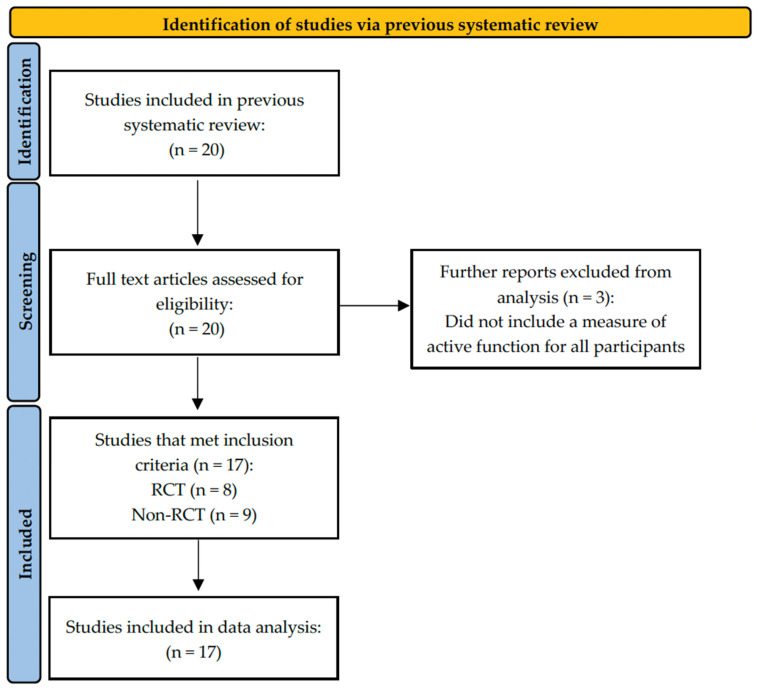
Modified PRISMA flow diagram for identification of studies to obtain articles for review inclusion.

**Table 1 toxins-17-00362-t001:** Study characteristics for active upper limb function outcome (*n* = 7).

Author	Study Design	Population	Sample (*n*)	Mean Age (y)	Active Function Outcome Measures	Strength Outcome Measures	Follow-Up Time Points
Giray 2020 [41]	RCT	Stroke	20	46	BBT SIS	MI	3/52 3/12
Lannin 2020 [44]	RCT	Stroke	139	61	BBT GAS QOL-EQ5D-OH QOL-EQ5D-SC	Grip-HHD	3/12
Lannin 2022 [43]	RCT	Stroke	140	61	BBT GAS QOL-EQ5D-OH QOL-EQ5D-SC	Grip-HHD	12/12
Lee 2018 [45]	Non-RCT	Stroke	15	45	ARAT-Total ARAT-Grasp ARAT-Grip ARAT-Gross ARAT-Pinch BBT QDASH	MRC Grip-HHD	2/52 6/52
Lim 2016 [46]	Non-RCT	Stroke	18	Sa: 63 Ch: 52	MBI ª	MRC	4/52
Pandyan 2002 [47]	Non-RCT	Stroke	14	57 ^	ARAT	Isometric Force Transducer Grip-GSM	4/52
Rousseaux 2002 [49]	Non-RCT	Stroke	20	54	FIM-UL NHPT 60 s NHPT 9 blocks RMA-UL	MRC	2/52 2/12 5/12

ª—Assessed upper and lower limb; ^—data supplied by author upon request; ARAT—Action Research Arm Test; BBT—Box and Block Test; Ch—Chronic; FIM-UL—Functional Independence Measure—Upper Limb; GAS—Goal Attainment Scale; GSM—Grip Strength Meter; HHD—Handheld Dynamometry; MBI—Modified Barthel Index; MI—Motricity Index; MRC—Medical Research Council scale; *n*—Sample; NHPT—Nine Hole Peg Test; QDASH—Quick Disabilities of Arm, Shoulder and Hand; QOL-EQ-5D OH—Quality of Life Euroqual-5D Overall Health; QOL-EQ-5D SC—Quality of Life Euroqual—5D Self-care; RCT—Randomised Controlled Trial; RMA-UL—Rivermead Motor Assessment Scale-Upper Limb; s—Seconds; Sa—Subacute; SIS—Stroke Impact Scale; y—years.

**Table 2 toxins-17-00362-t002:** Study characteristics for active lower limb function outcome (*n* = 10).

Author	Study Design	Population	Sample Size (*n*)	Mean Age (y)	Active Function Outcome Measure	Strength Outcome Measure	Follow-Up Time Points
Baricich 2019 [34]	RCT	Stroke	30	59	10MWT 2MWT	MRC	10/7 20/7 3/12
Bernuz 2012 [35]	Non-RCT	ISCI	15	43	10MWT-Gait Velocity 6MWT Timed Stair Climb	Isokinetic Peak Voluntary-Torque 60 °/s	4–6/52
Bollens 2013 [36]	RCT	Stroke	16	52.3	10MWT FAC FWC ABILOCO QOL-SF36 PH QOL-SF36 MH SATISPART-Stroke	MRC	2/12 6/12
Carda 2011 [37]	RCT	Stroke	69	Ta: 62 Ca: 65 St: 60	10MWT 6MWT FAC	MRC	3/52 3/12
Cinone 2019 [38]	RCT	Stroke	25	E: 56 C: 56	10MWT 6MWT	MI Isokinetic Dynamometry Peak-Torque 60 °/s	5/52 8/52
de Niet 2015 [39]	Non-RCT	HSP	25	E: 48 C: 46	10MWT-SS/FS TUG ABC	MRC QMA	4/52 18/52
Diniz de Lima 2021 [40]	RCT	HSP	55	43	10MWT-SS/FS SPRS	MRC	8/52
Hameau 2014 [42]	Non-RCT	Stroke	14	54	10MWT-SS/FS 6MWT TUG Ascend Stairs Descend Stairs	Isokinetic Dynamometer- MVC-Peak-Torque (5 test variations)	1/12
Rousseaux 2005 [48]	Non-RCT	Stroke	47	52	10MWT-SS/FS š 10MWT-Bf FAC š	MRC	2–3/52 2–3/12 5/12
Rousseaux 2007 [50]	Non-RCT	HSP	15	48 M	10MWT-SS/FS š FAC-Bf FAC š RMA-L+T	MRC	2–3/52 2–3/12 5/12

š—With Shoes and usual aids; °/s—degrees per second; 2MWT—Two-Minute Walk Test; 10MWT—10-Metre Walk Test; 6MWT-Six-Minute Walk Test; ABC—Activities-Specific Balance Confidence scale; Bf—Barefoot, no aids; C—Control; Ca—Casting; E—Experimental; FAC—Functional Ambulation Categories; FS—Fast Speed; FWC—Functional Walking Category; HSP—Hereditary Spastic Paraplegia; ISCI—Incomplete Spinal Cord Injury; M—Median; MI—Motricity Index; MRC—Medical Research Council Scale; MVC—Maximum Voluntary Contraction; *n*—Sample; QOL-SF36-PH—Quality of Life Short Form 36-Physical Health; QOL-SF36-MH—Quality of Life Short Form 36-Mental Health; QMA—Quantitative Muscle Assessment; RCT—Randomised Controlled Trial; RMA-L+T—Rivermead Motor Assessment Scale-Leg and Trunk; SATISPART(Stroke)—Instrument for the assessment of activity and participation following a stroke; SS—Self-selected; St—Stretching; Ta—Taping; TUG—Timed Up and Go Test; y—years.

**Table 3 toxins-17-00362-t003:** Upper limb strength and active function outcomes (*n* = 7).

ACTIVE FUNCTION		≤6/52 Weeks			>6/52 Weeks to ≤3/12 Months			>3 to ≤12/12 Months		
		Stronger	No Change	Weaker	Stronger	No Change	Weaker	Stronger	No Change	Weaker
		UPPER LIMB AGONIST STRENGTH								
	Improved		FF/Grip [45] EF/Grip [47] EF/WF [46]							WF [49]
	No change	Global [41]	FF/Grip [45] FF/Grip [45] EF/WF [46]		Global [41]	Grip [44]			Grip [43]	WF [49]
	Worse									
		UPPER LIMB ANTAGONIST STRENGTH								
	Improved	EE [46]	FE [45] EE [47] WE [46]					WE/FE [49]		
	No change		FE [45] FE [45] EE/WE [46]					WE/FE [49]		
	Worse									

EE—Elbow Extensors; EF—Elbow Flexors; FE—Finger Extensors; FF—Finger Flexors; WE—Wrist Extensors; WF—Wrist Flexors.

**Table 4 toxins-17-00362-t004:** Lower limb strength and active function outcomes (*n* = 10).

ACTIVE FUNCTION		≤6/52 Weeks			>6/52 Weeks to ≤3/12 Months			>3 to ≤12/12 Months		
		Stronger	No Change	Weaker	Stronger	No Change	Weaker	Stronger	No Change	Weaker
		LOWER LIMB AGONIST STRENGTH								
	Improved		Global [38] PF [48]	KE [35] PF [38] PF [39]		PF/Global [38] PF [48]			PF [39] PF [48]	
	No change		Global [38] PF [48]	KE [35] KE [42] PF [38]		HAd/PF [40] PF [36] PF [48] Global [38]	PF [38]		PF [48] [39] [36]	
	Worse									
		LOWER LIMB ANTAGONIST STRENGTH								
	Improved	DF [38]	DF [34] DF ^‡^ [37] DF [48] HAb/DF [50]		DF [38]	DF [34] DF ^‡^ [37] DF [48]			DF [48] HAb/DF [50]	
	No change	KF [42] DF [38]	KF [42] DF ^‡^ [37] DF [34] DF [48] HAb/DF [50]			DF ^‡^ [37] DF [36,38,48] HAb/DF [50]			DF [36,48] HAb/DF [50]	
	Worse									

^‡^—*p* value set at *p* < 0.02; DF—Dorsiflexors; HAb—Hip Abductors; HAd—Hip Adductors; KE—Knee Extensors; KF—Knee Flexors; PF—Plantarflexors.

## Data Availability

The original contributions presented in this study are included in the article/Appendix A. Further inquiries can be directed to the corresponding author.

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
