# Peer review of "The Impact of Strength Changes on Active Function Following Botulinum Neurotoxin-A (BoNT-A): A Systematic Review"

_toxins, 2025, doi:10.3390/toxins17080362_

Round 1
Reviewer 1 Report
Comments and Suggestions for Authors
This systematic review addresses an important clinical question and uses rigorous methodology overall. The authors provide a clear synthesis of the available evidence, identifying critical gaps and suggesting the need for standardized measures in future research.
However, I think it is very important to point out the following: The authors discuss concerns that BoNT-A may weaken muscles and potentially impact active function. However, they do not address that increasing the precision of BoNT-A injections (for example, through ultrasound guidance) is critical for minimizing the risk of muscle weakening. Ultrasound guidance allows for accurate targeting of spastic muscles and can help lower the required BoNT-A doses to the minimum necessary, reducing the potential for unintended weakness in surrounding muscles. This would be an important practical implication for clinical practice and may also partially address the uncertainty around muscle weakening in the included studies. Therefore, I recommend the authors expand their discussion to include the potential role of ultrasound-guided BoNT-A injections in achieving both precise dosing and optimal functional outcomes.
Please see this paper: Popescu MN, Căpeț C, Beiu C, Berteanu M. The Elias University Hospital Approach: A Visual Guide to Ultrasound-Guided Botulinum Toxin Injection in Spasticity: Part I—Distal Upper Limb Muscles. Toxins. 2025; 17(3):107. https://doi.org/10.3390/toxins17030107
Reviewer 2 Report
Comments and Suggestions for Authors
This systematic review addresses an interesting topic on the long term effects of Botulinum neurotoxin A on active function and strength. This systemic review is registered and follows the guidelines. This was an elaboration of a previous systematic review by Gill et al in 2024. Twenty studies were analyzed and 17 met the inclusion criteria. The Pedro scores for these 17 studies were fair ( mean score 18/28 ) and 9 out of 17 studies were not RCTs..
In the results they noted that in general, where active function outcomes improved, (25%) both upper and lower limb strength remained unchanged with a small proportion showing increased strength ( 15 %) and small percentage (10%) showing decreased strength.
Strength and active function of the upper limb (70% were mostly unchanged across all time points. When changes in the upper limb strength did occur, active function was mostly unchanged. For the lower limb, the trend was for lower limb weakness early post injection (90%) while active function remained unchanged (73%) or improved (27%) No studies reported weakness in the antagonist muscles or deterioration in active function at any time point.
The discussion summarizes the challenges to this systematic review and highlights the limitations.
The conclusion was that there was no clear relationship between a change in muscle strength and active function following BoNT-A injections to the upper and lower limbs for focal spasticity in adult-onset neurological conditions. The authors recommend that further studies regarding the use of Botox for spasticity should be precise, responsive, and include goal specific measurable outcomes. Using standardized outcome measures would assist in standardization of assessment in studies using BoNT-A for spasticity compared to alternative treatment or controls.
Majpr Concerns
Why were the methods reported after the results, discussion and conclusion?
The fair Pedro scores raise question about the validity of the findings.
The conclusion needs to be questions. The review did not provide enough consistent information to make a directionall conclusion. Rather, the conclusion is that a decision could not be made on the relationship of botox over time and its impact on strength and its relationship to function/quality of life.
The references should be consistently included by reference number and not date ( e.g. Gil on page 10 is reference 26 but here 2024 is in parentheses.
The findings reported by the systematic review by Gil should be provided in the introduction since the current review was an extension and potentially more complete than the systematic review by Gil.
Table 3. Upper extremity measurements. Not clear what the numbers meant in the brackets. Also very little information was provided at the 52 weeks to 3/12 months and 12/12 months. In the text it states measurements were not taken. Similarly little data was provided in the 12 month follow up. In Table 4, Lower Extremity Measurements., There were lots of measurements at 6/52 weeks but minimal measurements at 52 weeks/3 ½ months or 12/12 months. Again it is not clear what the numbers in the brackets are. However, both of these tables have so little data over time, one wonders whether it is necessary to include these tables.
No forest plots were calculated ( no meta-analysis). The authors stated that there was heterogeneity of study design, a lack of common outcome measurements of strength and active function and a lack of measurements over time. In the 17 studies. Thus it was not possible to conduct a meta-analysis. Unfortunately, this was the purpose of the systematic review. Thus, one has to question whether this review Is worth publishing. What does it add that the Gil systematic review in 2024 did not include?
The supplementary files provide more data but still not very meaningful relative to the systematic review. However, Table 2 in the supplementary files provides detailed information on each of the 17 studies. If this manuscript is published , I would suggest that this table be included in the primary file to enable the reader to see more detail about the studies.
Reviewer 3 Report
Comments and Suggestions for Authors
Toxins 3699136
This manuscript is a review of a review. There is complete reliance on the previous review by Gill et al 2024. As such, the present review is only of the 20 publications selected by Gill et al.
There is no mention anywhere of the possible effect of dose. Success or failure of BoNT-A treatments can be highly dependent on the dose used. I recommend that the authors revise their manuscript to include discussion of the potential effect of dose and the variability of doses used in the studies they have analysed. To note, any so-called “dose ratio” to compare the effect of different products should also be discussed. At present, the analyses carried out have an underlying (and not mentioned) assumption that all studies use the same dose, which cannot be correct.
Title
I recommend a small modification to the title to read more correctly in English
The impact of strength changes on Active Function following Botulinum Neurotoxin-A (BoNT-A) injection: A Systematic Review
Abstract
Line 7 “further” should be removed
The abstract needs to be clear earlier on that only effects on upper and lower limb were investigated
Citation [24] To note that these authors used a very high dose of BoNT-A that is not clinically relevant
Citation [26] These authors conclusion was as follows:
Overall, the impact of BoNT-A on muscle strength remains inconclusive.
This is not the same as stated on lines 45-46
The Abstract should make clear that the present manuscript is a review of a review, namely Gill et al 2024, and not a de novo review. The searches of the databases described were carried out by Gill et al as described in Section 6.2.
Methods
Lines 277 -279 mention BOTH systematic reviews, but only one registration number is given. Only the review by Gill et al is included under this registration number.
Lines 324-326 An indication should be provided of how successful contacts with authors, for the studies examined, was.
Line 338 A citation for these guidelines is needed
Line 349 More detail is required on which summary statistics were used
Results
Figure 1 Identification of studies via previous studies
This should be reworded
Line 242 Citation [75] is related to use of BoNT-A in aesthetics. This is not a good inclusion given the very large difference in doses used in aesthetics compared to therapeutic UL or LL applications and the significant differences in muscle organisation, with multiple neuromuscular junctions in facial muscles throughout their length.
Reviewer 4 Report
Comments and Suggestions for Authors
This is a well-executed and valuable secondary analysis of a systematic review. The research question is novel and of high clinical relevance, moving beyond the simple effect of BoNT-A on impairment (strength) to its impact on function.
In title secondary analysis should be specified
Abstract
Observation: The abstract currently states that eight databases were searched. This is true for the original review but not for this specific analysis. It would be more transparent and accurate to state from the outset that this is a secondary analysis. The methods section of the abstract states that "Eight databases...were searched." This is misleading. This specific study did not perform a search; it screened the results of a previous search.
Please revise the methods description in the abstract to clarify its relationship with the previous review.
The phrase "...reporting 308 outcomes..." is slightly confusing in the context of an abstract. It is not immediately clear what an "outcome" refers to (e.g., a time point, a specific measurement, a subscale). I recommend removing the mention of "308 outcomes" from the abstract.
The "Key Contribution" statement begins by stating, "There was no clear evidence that BoNT-A causes muscle weakness..." While true according to the authors' previous work, this is the conclusion of the first paper, not the primary finding of this paper. The key contribution here is the relationship between strength and function.
Introduction
The manuscript cites the previous review and notes that it "did not report the implications of these changes on active function." This link is good, but it could be made more explicit to strongly justify the current paper's existence. Please expand slightly on this sentence to more clearly frame the knowledge gap this paper fills.
Methods
The definition of "active function" is provided here. As this is the central outcome of the entire review, its definition is critical. To improve the flow and emphasize its importance, I recommend moving this operational definition of "active function" to the end of the Introduction, immediately before the stated aim of the study.
The third additional inclusion criterion is that "The measure of active function demonstrated known clinometric properties." The term "known" is slightly subjective.
Please briefly specify how this was determined.
Results
The text reports the overall numbers but does not walk the reader through a specific example from the complex tables (Table 3 and 4).
Please add a sentence in each section (upper and lower limb) to guide the reader.
Discussion
The discussion correctly identifies the main finding—the disconnect between strength and function—and points to methodological issues like outcome measure responsiveness. However, it could further explore the potential neurophysiological and clinical reasons for this disconnect.
Please add a paragraph to the discussion that speculates on why function might improve even when strength does not. The primary role of spasticity reduction: BoNT-A's main effect is reducing spasticity. This may "unmask" a patient's existing, usable strength, allowing for better movement without any actual increase in force-generating capacity. The functional gains may be due to improved coordination, reduced antagonist co-contraction, or better selective motor control, none of which are captured by a simple strength measure. Furthermore, the potential imapact on Unpleasant sensations (doi: 10.3390/jcm13061720.). As noted in the limitations, therapies like task-specific training, which were often co-interventions, directly target function and may be the primary driver of functional gains.
Round 2
Reviewer 2 Report
Comments and Suggestions for Authors
The authors have responded to the reviewers and I believe the manuscript is ready for acceptance.
Reviewer 3 Report
Comments and Suggestions for Authors
I have reviewed the changes made by the authors in response to my initial comments on their manuscript. I have two continuing concerns.
- I highlighted in my initial review that the original manuscript said nothing about the effect of dose on the clinical results. This is one of the most important aspects of treatment with BoNT-A. The authors addressed this comment by a small change in lines 282-286. This minor change on such a major issue is disappointing. I would have expected much more to be included in the manuscript on the subject and very much earlier in the text, also in the Abstract. The entirety of their analyses could be heavily biased by differences in dose and products used. I recommend that the authors revise their manuscript accordingly, as I initially proposed.
- I have checked again the PROSPERO data base and can still only see the original Gill et al publication included under the number the authors have provided https://www.crd.york.ac.uk/PROSPERO/view/CRD42022315241 . This must be addressed.
Reviewer 4 Report
Comments and Suggestions for Authors
The revised version addressed the main concerns.
